# Virtual Screening and Biological Activity Evaluation of New Potent Inhibitors Targeting Hexokinase-II

**DOI:** 10.3390/molecules27217555

**Published:** 2022-11-04

**Authors:** Ruijuan Liu, Xuewei Liu

**Affiliations:** 1College of Physical Education, Northwest Normal University, Lanzhou 730070, China; 2Department of Pharmacy, Jiangsu Food and Pharmaceutical Science College, Huaian 223003, China

**Keywords:** Hexokinase-II, molecular docking, inhibitor, virtual screen, small molecule

## Abstract

Hexokinase-II (HK-II), the rate-limiting step enzyme in the glycolysis pathway, expresses high levels of cancer cells compared with normal cells. Due to its pivotal role in the different aspects of cancer physiology including cellular proliferation, metastasis, and apoptosis, HK-II provides a new therapeutic target for cancer therapy. The structure-based virtual screening targeting HK-II was used to hit identifications from small molecule databases, and the select compounds were further evaluated in biological assays. Forty-seven compounds with the lowest binding energies were identified as potential HK-II inhibitors. Among them, nine compounds displayed the highest cytotoxicity to three different cancer cells. Based on the mechanism study, compounds 4244-3659 and K611-0094 showed an obvious inhibitory effect on the HK-II enzyme. This study identified two potential inhibitors of HK-II and can be helpful for developing potential drugs targeting HK-II in tumor therapy.

## 1. Introduction

Glucose metabolism is the intracellular biochemical transformation process from glucose to pyruvate, which generates extracellular adenosine triphosphate (ATP) [1]. Without oxygen consumption, pyruvate is reduced to lactate both in normal and diseased cells. Pyruvate prefers to generate lactate rather than enter the TriCarboxylic Acid (TCA) cycle in cancer cells, even under conditions of adequate oxygen [2]. Most cancer cells are heavily dependent on aerobic glycolysis for energy supply instead of mitochondrial oxidative phosphorylation [3,4]. Thus, targeting aerobic glycolysis may be a promising alternative for cancer therapy [5].

Hexokinase-II (HK-II) catalyzes the rate-limiting step in glycolysis and promotes the generation of glucose-6-phosphate (G6P) from glucose and ATP [6]. HK-II bound to the Voltage Dependent Anion Channel (VDAC) catalyzes the phosphorylation of glucose by utilizing ATP generated in mitochondria and suppresses the release of gap junction membrane proteins, thereby leading to the inhibition of apoptosis and the induction of tumor cell proliferation and growth [7,8,9,10].

Early studies have documented that a high expression level of HK-II was expressed in cancer cells [11,12]. The knockdown of the expression of HK-II contributes to the enhancement of the oxidative phosphorylation metabolism and the inhibition of aerobic glycolysis. Thus, the binding of HK-II to VDAC and the elevated HK-II expression contribute to the Warburg effect in cancer cells, which implies that HK-II might act as a therapeutic target for novel anticancer drug screening and design [13,14].

HK-II represents a promising approach to target cancers, but very few HK-II inhibitors have been reported. The anti-diabetic drug, metformin, shows a weak inhibitory effect against HK-II (IC50 > 10 mM) [15,16,17]. Notably, 2-Deoxyglucose (2-DG), an inhibitor of glycolysis, shows no significant therapeutic effect of a single agent at the dose range from 500 mg/Kg to 2000 mg/Kg in mice [18,19]. Additionally, 3-Bromopyruvate (3-BrPA) exhibits a moderate inhibitory action against HK-II, but the strong off-target and serious side effects limit its application [20]. In recent years, Liu et al. identified two classes of small-molecule HK2 inhibitors, i.e., benserazide and (*E*)-*N*′-(2,3,4-trihydroxybenzylidene) arylhydrazides, by the structure-based virtual screening of the ZINC database [21,22,23]. Although €-4-Nitro-*N*′-(2,3,4-trihydroxybenzylidene), with an enzyme activity of 0.53 ± 0.13 μM, was recently designed and synthesized, its possible mechanism of action and the accompanying cellular effects have not been fully clarified [23]. Because of their unsatisfactory therapeutic effects, obvious side effects, and unclear mechanisms, developing novel HK-II inhibitors with high efficiency and low toxicity is still an urgent matter [24].

Virtual screening has been widely used to screen large libraries of compounds and to identify those structures likely to bind to a drug target. In the present work, we conducted molecular docking-based virtual screening and screening strategies to find out a new series of HK-II inhibitors. The cell viability and HK-II activity were detected to further screen the small molecules obtained by virtual screening. Two potential inhibitors, compounds 4244-3659 and K611-0094, were found to bind to HK-II and exhibit an effective inhibitory effect on the HK-II enzyme.

## 2. Results and Discussion

### 2.1. Virtual Screening and Initial Hit Evaluation

The workflow of the structure-based virtual screening is shown in Figure 1. Based on the results of virtual screening, forty seven compounds were selected for cytotoxicity tests. Initially, we adopted the MTT assay to screen the cytotoxicity of all compounds toward three different types of cancer cell lines (HeLa, HepG 2 and A549). The physical characteristics and docking results of compounds and the concentrations with the IC_50_ values (inhibit the cell proliferation to 50% of the control) below 100 μM are summarized in Table 1 and Table 2, respectively. Some compounds showed a very promising cytotoxicity, such as K788-8853, K263-0793, and the three compounds selected from the Specs database. Next, we chose these nine compounds for further follow-up studies. The chemical structures of the selected compounds are depicted in Figure 2.

### 2.2. Hexokinase-II Activity

Since nine compounds display the highest cytotoxicity to three different cancer cells, we then verified the effects of the compounds on Hexokinase-II activity. The Hexokinase-II enzyme was incubated with various concentrations (from 1 to 128 μM) of every compound, and six of these compounds showed varying degrees of inhibition. The obtained results were given in Table 3. For the tested compounds, compounds 4244-3659 and K611-0094 have obvious inhibitory effects on Hexokinase-II enzyme (Figure 3), with IC_50_ readings of 17.5 ± 1.4 μM and 45.0 ± 1.1 μM, respectively.

### 2.3. Structural Analysis of the Binding of Hit Molecules to HK-II

In Figure 4, we compared the protein-ligand binding modes of the two hit molecules to HK-II (PDB ID: 2NZT), K611-0094 and 4244-3659. Both hit molecules can be closely bonded to the receptor protein HK-II through hydrogen bond interaction and steric hindrance. Compared with hit compound K611-0094, 4244-3659 has a higher ratio of acceptor/donor of HB, and less steric hindrance (Table 1 and Figure 4).

Then, we also conducted a further analysis of the binding mode of compound 4244-3659 to HK-II (Figure 5). The compound 4244-3659 molecule consists of two main rings, the hydroquinone ring (ring A) and the benzo butyl lactam ring (ring B). The hydroxyl group at the meta-position on ring A as a hydrogen bond donor forms a hydrogen bond with the side-chain carbonyl oxygen of Asp209. In addition, protons on Asn235 and Asn208 form two hydrogen-bonding interactions with oxygen atoms on the hydroxyl group. Meanwhile, the hydroxyl group at the ortho position on ring A forms a hydrogen bond with the carboxyl oxygen atom at Glu294. The phenyl ring in ring A forms a Van der Waals interaction with the Phe156 and Pro157 nonpolar amino acids. The strong hydrogen bonding and hydrophobic interactions allow ring A to bind to the center of the receptor active site pocket stably. The benzene ring stretches to the edge of the pocket as ring B turns outwards, which avoid the collision between ring B and the surrounding amino acids, while it is beneficial to the formation of hydrogen bonds between the hydroxyl oxygen atoms on the butyrolactam ring and hydrogen atoms on the Gly233 skeleton, as well as the hydrogen atoms on hydroxyl and the side chain oxygen atoms on Thr232. It was obtained from the analysis that more hydrogen bonds existed between the 4244-3659 molecule and the amino acids around the receptor active pocket. Overall, the higher ratio of the acceptor/donor of HB increases the ligand and protein binding ability, and the hydrogen bonding interaction stabilizes ligand receptor binding. The hit molecule 4244-3659 can be used as a lead compound for the development of HK-II inhibitors.

## 3. Materials and Methods

### 3.1. Docking-Based Virtual Screening

The virtual screening workflow was performed using Schrödinger software (Schrödinger, LLC, New York, NY, USA; Schrödinger, 2015). The crystal structure of human HK- II (PDB ID: 2NZT) was obtained from Protein Data Bank and prepared using the Protein Preparation Wizard in Schrödinger. There are two compounds co-crystallized with HK-II in PDB 2NZT, 6-*O*-phosphono-beta-d-glucopyranose and alpha-d-glucopyranose. The combination pocket consisted of residue Thr88, Asp89, Thr172, Lys173, Asn208, Asp209, Thr32, Asn235, Glu260, Glu294, Asp413, Ser415, etc., in the original complex, which was identified by the docking parameter file. All compounds from ChemDiv and Specs libraries (containing 190,000 chemicals) were preprocessed using the LigPrep module of Schrödinger. The ChemDiv database contains more than 160,000 compounds and has specific structural motives grouped in screening sets like mimetics compounds, spiro compounds, and cyclic compounds. The Specs database contains nearly 300,000 compounds and provides unique and novel building blocks. The Epik algorithm in the work of Schrödinger was used to predict the ionized states, tautomers, and stereoisomers at pH 7.0 [25]. All compounds were docked using the standard extra precision of the Glide module, and then Induced Fit Docking was carried out to redock the top compounds. The high-scoring docked molecules were identified based on the results of the docking calculation. Then, the structure and binding mode of candidate molecules were evaluated using the clustering protocol integrated into the Canvas module of Schrödinger. A Tanimoto coefficient cutoff value of 0.5 was used in the analysis of clustering.

Compounds satisfying most of Lipinski’s rules, which were known as filters of drug-likeness, displayed better pharmacokinetic properties and bioavailability. During the sieving process, we used Lipinski rules to filter molecules and calculated the physicochemical properties of each compound to remove the molecules with active groups and reactive groups. After docking through all the three precision modes of Glide HTVS (High Throughput Virtual Screening), SP (Standard Preparation), and XP (Extra Precision), 4000 compounds with top Glide docking scores were obtained from the ChemDiv and Specs libraries. Then, the MM-GBSA values of these top-ranked complexes were calculated to predict the ligand-binding affinities. Following MM-GBSA treatment, the 4000 molecules were clustered into 100 classes based on the K-Means clustering in Canvas. According to the procedure described above, forty-seven compounds were finally selected for the evaluation of the HK-II kinase inhibition assay.

The molecular docking calculation was implemented using the Schrödinger 2015 package to determine how compounds bound to HK-II. The processing of the HK-II structure was performed using Schrödinger’s Protein Preparation Wizard. The structures of the compounds were then optimized by LigPrep using an OPLS-2005 force field. The active binding site was defined by the docking grid box. The molecules were docked into the HK-II binding site using the Glide docking program in the standard precise mode. The top-scoring pose was selected for further analysis.

### 3.2. Chemicals and Enzymes

Penicillin, 3-(4, 5-Dimethylthiazol-2-yl)-2,5-diphenyltetrazolium bromide (MTT), and streptomycin were obtained from Amresco (Solon, OH, USA). Antibodies of HK2 (Code: A0994) were purchased commercially from ABclonal (Wuhan, China). Dulbecco’s modified Eagle’s medium (DMEM), DMSO, Fetal bovine serum (FBS), and 3-(3-Cholamidopropyl) dimethylammonio-1-propanesulfonate (CHAPS) were obtained from Sijiqing (Hangzhou, China). All other purchased chemical reagents were of analytical grade and were used without further purification. All compounds were purchased from Topscience (Shanghai, China) and dissolved in dimethyl sulfoxide (DMSO) until the concentrations reached 20 mM. All of the reported concentrations of DMSO, unless specially notified, did not exceed 0.1% (*v*/*v*).

### 3.3. Cell lines and Culture Conditions

HeLa, Hep G2, and A549 cells were purchased from the Shanghai Institute of Biochemistry and Cell Biology, Chinese Academy of Sciences. All cells were cultured in DMEM with 100 units/mL penicillin/streptomycin, 2 mM glutamine, and 10% FBS in a carbon dioxide atmosphere at 37 °C.

### 3.4. Cytotoxicity Assays

Cytotoxicity was quantified using a standard MTT assay, which is widely used in evaluating cell proliferation and screening for anticancer drugs. Briefly, following 7 × 10^3^ cells incubating in 96-well plates for 48 h filled with DMEM medium supplemented with 10% fetal bovine serum (FBS) containing different concentrations of samples, 10 μL/well of MTT (5 mg/mL) was added. After 4 h of incubation at 37 °C, 100 μL of extraction buffer containing 5% iso-butanol, 0.1% HCl, and 10% SDS was added, and then the cells were incubated overnight. As controls, cells were treated with 0.1% (*v*/*v*) DMSO alone. The cell viability was calculated by the absorbance (570 nm) of each well obtained using a microplate reader (Thermo Scientific Multiskan GO, Vantaa, Finland).

### 3.5. Hexokinase-II Activity

The Hexokinase-II activity was tested via a Hexokinase-II inhibitor screening kit (Abcam, Cambridge, UK, ab211114), and all experiments were carried out following the manufacturer’s instructions. Briefly, 50 μL test compound (from 1 to 128 μM) mixed with 5 μL Hexokinase-II enzyme solution was incubated for 5 min at 25 °C, and then 45 μL substrate mix was added to each well of 96-well plates. HK assay buffer was treated as negative control, bromopyruvic acid was used for positive controls, and DMSO was used as a solvent control. The value of OD at 450 nm at 15 min and 25 min was recorded. IC_50_ values were calculated in GraphPad Prism 5.

### 3.6. Statistical Analysis

All experiments were repeated at least three times, and all analytical data are presented as means ± SE. The statistical significance of comparisons between groups of data was determined with Student’s t-test. Differences between datasets were assessed by analysis of variance (ANOVA). A *p* value < 0.05 was considered to be statistically significant.

## 4. Conclusions

High levels of HK-II expression may be a possible cause of the Warburg effect and might be a prognostic marker for cancer. Thus, HK-II should be highlighted as a valid anticancer drug target. In this research, we employed a molecular docking-based virtual screening to identify novel and potential HK-II inhibitors from two compound libraries. The cell viability and HK-II activity were performed on these selected compounds, nine of them showed promising cytotoxicity to three different cancer cells, and two compounds with effective inhibitory on HK-II were selected as hit compounds. Further analysis of the binding mode of compound 4244-3659 to HK-II showed that hydrogen bonding plays a leading role in the stable binding of the receptor to the ligand. Our study led to the identification of two novel HK-II inhibitors, which have potential as lead compounds for therapeutic agents in cancer.

## Figures and Tables

**Figure 1 molecules-27-07555-f001:**
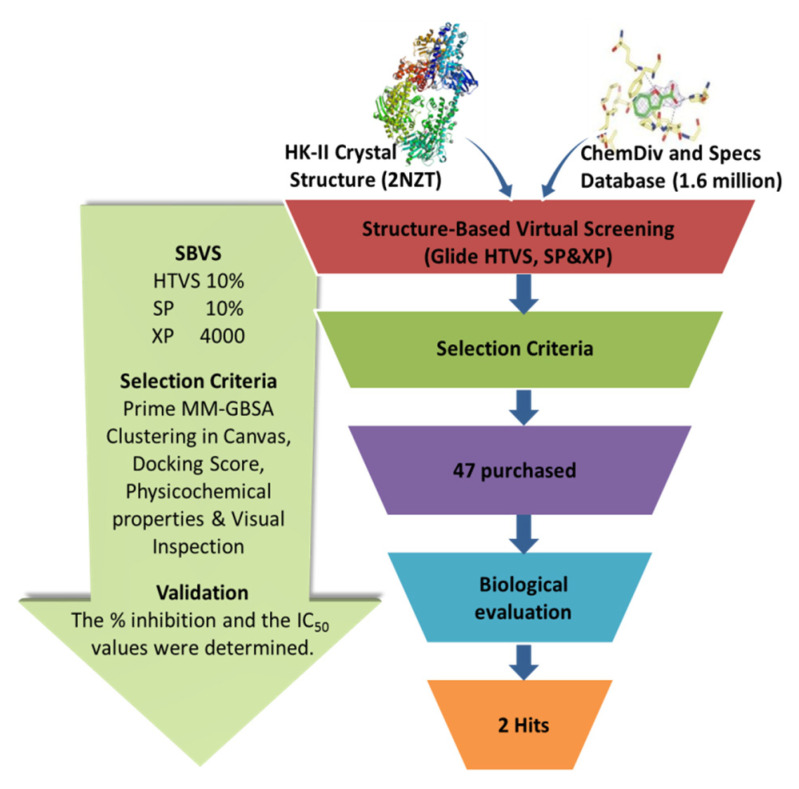
The virtual screening workflow.

**Figure 2 molecules-27-07555-f002:**
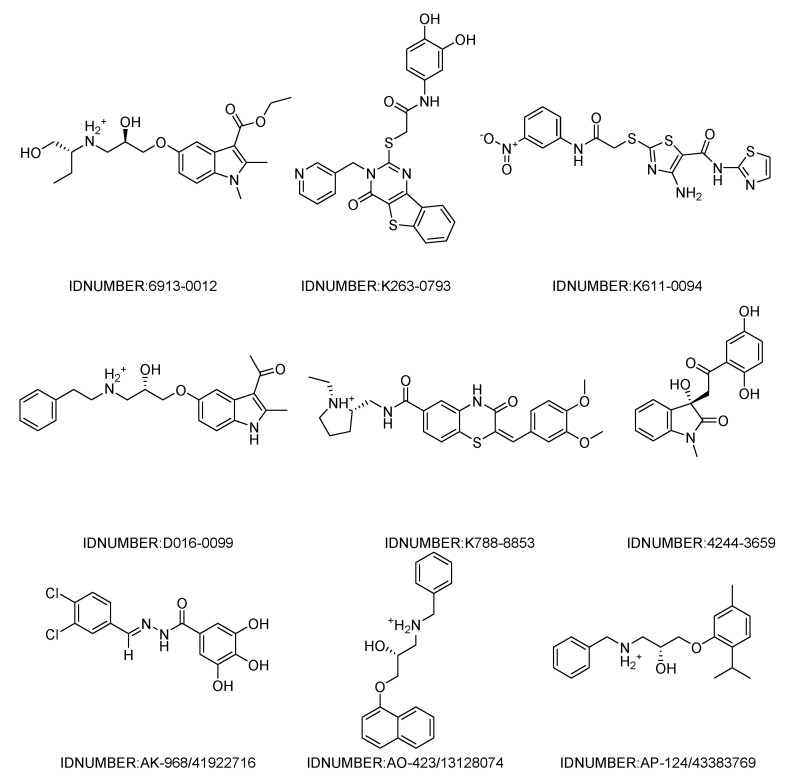
Chemical structures of the nine compounds identified by virtual screening.

**Figure 3 molecules-27-07555-f003:**
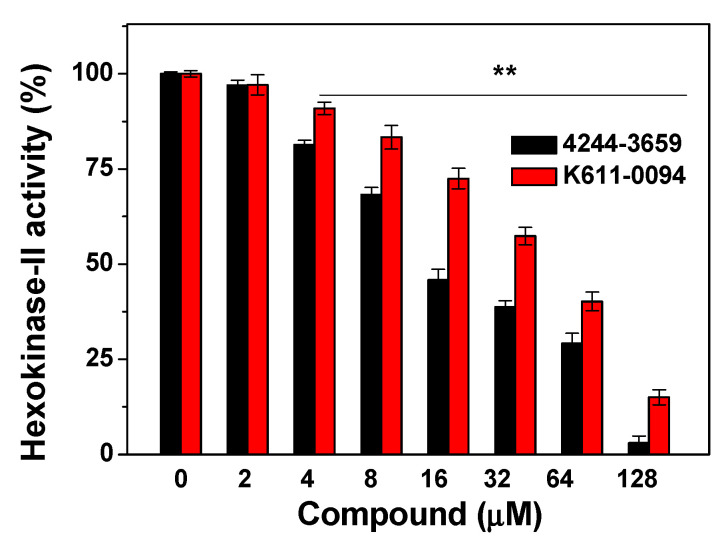
Inhibition of compounds 4244-3659 and K611-0094 on Hexokinase-II activity. After Hexokinase-II enzyme was incubated with various concentrations (from 1 to 128 μM) of compound, Hexokinase-II activity was tested via a Hexokinase-II inhibitor screening kit. All data were expressed as mean ± SE from three independent experiments. *** p* < 0.01 vs. the control groups.

**Figure 4 molecules-27-07555-f004:**
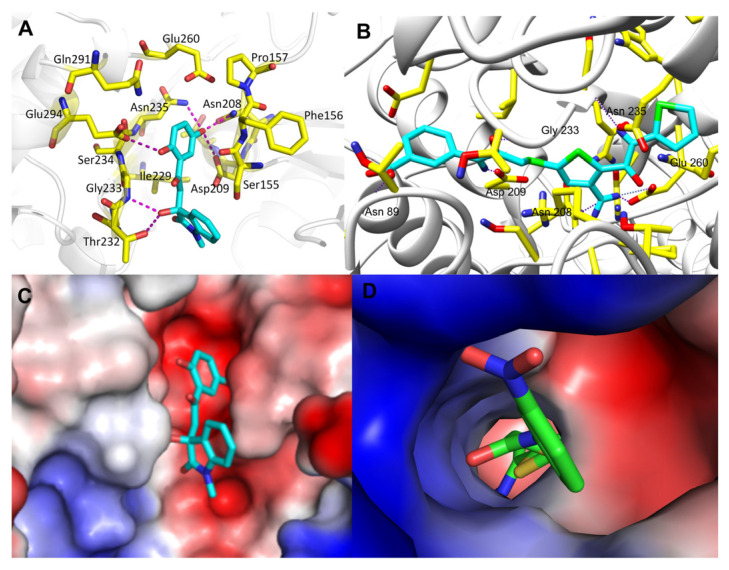
The binding mode analysis of hit compounds and HK-II. (**A**) 4244-3659 and protein interaction of the three-dimensional binding mode map. (**B**) K611-0094 and protein interaction of the three-dimensional binding mode map. (**C**) Electrostatic interaction between 4244-3659 and protein interaction map. (**D**) Electrostatic interaction between K611-0094 and protein interaction map.

**Figure 5 molecules-27-07555-f005:**
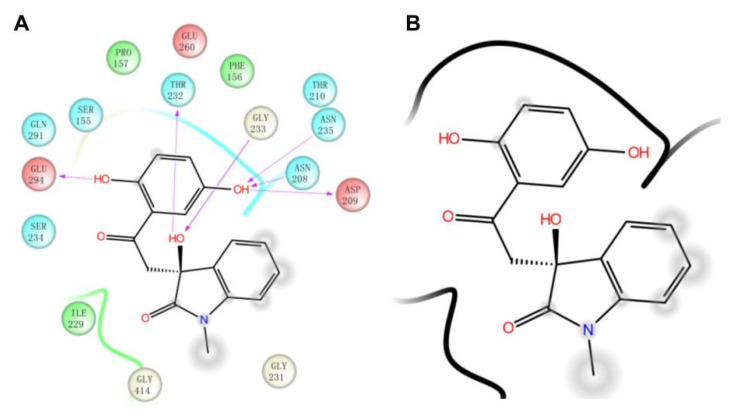
The binding site of HK-II(PDB ID: 2NZT) with compound 4244-3659. (**A**) The two-dimensional binding pattern of the interaction between 4244-3659 and protein. (**B**) 4244-3659 and protein interaction ligand molecular structure diagram.

**Table 1 molecules-27-07555-t001:** Information of nine selected compounds.

Database		acceptHB ^a^	donorHB ^b^	Docking Score ^c^	Prime MM-GBSA (kcal/mol)
Chemdiv	6913-0012	7.65	3	−8.5291	−46.0649
	K263-0793	9.5	3	−8.0771	−60.5486
	K611-0094	8.5	3	−7.8236	−52.1084
	D016-0099	4.95	2	−7.3413	−43.2341
	K788-8853	8.5	2	−7.2758	−47.5018
	4244-3659	5.25	1	−7.1817	−42.5366
Specs	AK-968/41922716	4.75	4	−7.0789	−47.2578
	AO-423/13128074	3.95	2	−6.6181	−38.1110
	AP-124/43383769	3.95	2	−6.3516	−38.7376

^a^ Number of acceptor of hydrogen bond; ^b^ Number of donor of hydrogen bond; ^c^ Docking scores of compound with HK-II.

**Table 2 molecules-27-07555-t002:** Toxicity for each compound predicted by docking.

Database		IC_50_ (μM) ^a^		
		Hela	HepG 2	A549
Chemdiv	6913-0012	51.9 ± 1.5	59.0 ± 0.2	62.6 ± 0.7
	K263-0793	22.4 ± 1.4	22.1 ± 0.9	27.1 ± 1.4
	K611-0094	47.8 ± 0.6	57.9 ± 1.7	48.8 ± 1.2
	D016-0099	48.5 ± 1.2	44.2 ± 0.6	50.4 ± 0.3
	K788-8853	15.1 ± 0	12.0 ± 0.5	10.7 ± 0.5
	4244-3659	62.1 ± 2.0	62.5 ± 2.1	54.6 ± 2.6
Specs	AK-968/41922716	28.5 ± 1.9	40.0 ± 0.2	31.0 ± 0
	AO-423/13128074	27.7 ± 0.6	30.0 ± 1.4	38.6 ± 0.8
	AP-124/43383769	25.1 ± 0.9	23.3 ± 0	36.5 ± 0.4

^a^ toxicity data of the nine compounds.

**Table 3 molecules-27-07555-t003:** Inhibitory effect of compounds on HK-II enzyme activity.

Database		EC_50_ (μM)
Chemdiv	6913-0012	151.7 ± 1.1
	K263-0793	156.3 ± 1.1
	K611-0094	45.0 ± 1.1
	4244-3659	17.5 ± 1.4
Specs	AO-423/13128074	153.8 ± 1.2
	AP-124/43383769	103.8 ± 1.1

## Data Availability

Not applicable.

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
