# Peer review of "Virtual Screening and Biological Activity Evaluation of New Potent Inhibitors Targeting Hexokinase-II"

_molecules, 2022, doi:10.3390/molecules27217555_

Round 1

Reviewer 1 Report

In their submitted manuscript, Liu et al. evaluate virtually screened
biologically active inhibitors of Hexokinase-II.
Although the manuscript is well written, this referee has some
concerns on the reliability of the assumption that HK-II is a
suitable target for cancer therapy due to a strong
cross-reactivity of the applied inhibitors with HK-II in healthy
cells. This referee strongly assumes that HK-II is linked to the
Glucose-related pathways and that HK-II expression levels only
play a role in distinct cancer types such as breast and colon
cancer. The cell lines selected by the authors do not reflect that
correlation but they use HeLa, Hep G2 and A549 cells, which rather
shows that the applied inhibitors are strongly cross-reactive and
rather are not applicable as efficient therapeutics. A distinct
difference in the affinities between mutageneous HK-II in a
cancer-type and healthy HK-II might be a reasonable issue to
publish. This study, in more detail, only shows that the
compounds available suppressed HK-II relevant pathways which
induces cytotoxicity. That is not an indicator that the presented
compounds are relevant as tumor suppressors.

However, the study in its form is well-suited as a model and might
be of use for the community as I think that the data is relevant
for other scientists. Therefore, I recommend to publish this
article.

Author Response

We thank you for your helpful comments.

Reviewer 2 Report

The paper describes virtual screening and testing of small molecules against Hexokinase-II, a possible cancer target.

The paper is very brief and could do more to interrogate the choice of approach and the results. The computational methods need revision/clarification, as there are some approaches mentioned in the Results that are not mentioned in the Methods. A combined Results And Discussion section is provided, but no discernible discussion is present. The authors should put their findings in greater context among the existing literature, to give the reader a better sense of why this work is interesting/important. Incorporating some of the below suggestions will assist with this, but the authors should consider a separated Discussion section that expands on this context.

- Noting that virtual screening at this target has already been performed using the ZINC library, which is more or less an exhaustive chemical library, the authors should clarify the particular advantages of using the ChemDiv and Specs library over ZINC, with reference to the size and chemical space (e.g., fragments, drug-like molecules, lead-like molecules) covered by each library, either in the introduction or in section 2.1.

- p2 line 69: please mention the ligand co-crystallised with HK-II in PDB 2NZT.

- Is it necessary to separate section 2.6 from 2.1? These could be combined into a single docking-based virtual screening section. It should not have been necessary to repeat the docking for given ligands following the virtual screening as applied.

- Section 3.1 reads more as methods than results, and appears to cover the methods with greater clarity than 2.1. It also includes mention of additional methods employed, specifically, Prime MM-GB/SA. Suggest to combine lines 68-74 of 2.1 with all of section 3.1 and to include the revised section in the methods.

- Section 3.2 is better retitled as “Virtual screening and initial hit evaluation”.

- Section 3.4 is not really a molecular docking study, but a “Structural analysis of the binding of hit molecules to HK-II” and should be retitled as such. Further, this section should be extended/revised in several ways:

- Both hit molecules identified from this study should be examined here.

- At a minimum, the molecule co-crystallised with HK-II should be examined here, and the interactions made by this and the hit molecules from this study compared. If other molecules have been co-crystallised with HK-II, they should also be examined and compared here.

- It is stated that “the hydrogen bonding plays a leading role in the stable binding of the receptor to the ligand”, but it is not actually possible to infer this from merely structural examination – however, it may be possible to infer this from either/both of the Glide scoring results and the Prime MMGBSA results. The authors should support this statement by examining how the hydrogen bonding/electrostatics components are contributing to the overall Glide Energy (specifically, does Glide Ecoul account for a large percentage contribution to Glide Energy?) and/or Prime MMGBSA energy (specifically, is MMGBSA dG Bind Coulomb + MMGBSA dG Bind GB negative, and if so, does that sum account for a large percentage contribution to MMGBSA dG Bind?). 

Author Response

Major Points:

  1. Noting that virtual screening at this target has already been performed using the ZINC library, which is more or less an exhaustive chemical library, the authors should clarify the particular advantages of using the ChemDiv and Specs library over ZINC, with reference to the size and chemical space (e.g., fragments, drug-like molecules, lead-like molecules) covered by each library, either in the introduction or in section 2.1.

Reply: According to the reviewer’s suggestion, we have clarified the particular advantages of using the ChemDiv and Specs library in section 2.1.

  1. p2 line 69: please mention the ligand co-crystallised with HK-II in PDB 2NZT.

Reply: According to the reviewer’s suggestion, we have supplemented the ligand co-crystallised with HK-II in PDB 2NZT.

  1. Is it necessary to separate section 2.6 from 2.1? These could be combined into a single docking-based virtual screening section. It should not have been necessary to repeat the docking for given ligands following the virtual screening as applied.

Reply: According to the reviewer’s suggestion, we have combined these two sections into a single docking-based virtual screening section.

  1. Section 3.1 reads more as methods than results, and appears to cover the methods with greater clarity than 2.1. It also includes mention of additional methods employed, specifically, Prime MM-GB/SA. Suggest to combine lines 68-74 of 2.1 with all of section 3.1 and to include the revised section in the methods.

Reply: According to the reviewer’s suggestion, we have combined section 3.1 to the revised methods section.

  1. Section 3.2 is better retitled as “Virtual screening and initial hit evaluation”.

Reply: According to the reviewer’s suggestion, we have retitled section 3.2.

  1. Section 3.4 is not really a molecular docking study, but a “Structural analysis of the binding of hit molecules to HK-II” and should be retitled as such. Further, this section should be extended/revised in several ways: Both hit molecules identified from this study should be examined here. At a minimum, the molecule co-crystallised with HK-II should be examined here, and the interactions made by this and the hit molecules from this study compared. If other molecules have been co-crystallised with HK-II, they should also be examined and compared here.

Reply: Thanks for the reviewer’s comments. We have retitled section 3.4 and supplemented the structural analysis of the binding of the hit molecule identified to HK-II.

  1. It is stated that “the hydrogen bonding plays a leading role in the stable binding of the receptor to the ligand”, but it is not actually possible to infer this from merely structural examination – however, it may be possible to infer this from either/both of the Glide scoring results and the Prime MMGBSA results. The authors should support this statement by examining how the hydrogen bonding/electrostatics components are contributing to the overall Glide Energy (specifically, does Glide Ecoul account for a large percentage contribution to Glide Energy?) and/or Prime MMGBSA energy (specifically, is MMGBSA dG Bind Coulomb + MMGBSA dG Bind GB negative, and if so, does that sum account for a large percentage contribution to MMGBSA dG Bind?). 

Reply: According to the reviewer’s suggestion, we have supplemented the number of Hydrogen bond and the Prime MMGBSA results and reorganized Table 1 and 2.

We thank you for handling the manuscript, and your helpful comments, both of which have greatly improved the quality of the manuscript. Having answered these queries, we now expect that the revised manuscript may be considered for publication.

Round 2

Reviewer 2 Report

Thank you, comments have been satisfactorily addressed.

Please review English/spelling in additions (e.g., section 2.1, "structural motives" likely should be be "structural motifs").